# Volume-Independent Sodium Toxicity in Peritoneal Dialysis: New Insights from Bench to Bed

**DOI:** 10.3390/ijms222312804

**Published:** 2021-11-26

**Authors:** Silvio Borrelli, Luca De Nicola, Ilaria De Gregorio, Lucio Polese, Luigi Pennino, Claudia Elefante, Alessandro Carbone, Tiziana Rappa, Roberto Minutolo, Carlo Garofalo

**Affiliations:** Unit of Nephrology, Department of Advanced Medical and Surgical Sciences, University of Campania “Luigi Vanvitelli”, 80138 Naples, Italy; luca.denicola@unicampania.it (L.D.N.); ila.degre.idg@gmail.com (I.D.G.); lucio.polese@live.com (L.P.); luigipennino24@gmail.com (L.P.); c.elefante91@gmail.com (C.E.); carbone448@gmail.com (A.C.); ra.tiziana@gmail.com (T.R.); roberto.minutolo@unicampania.it (R.M.); carlo.garofalo@unicampania.it (C.G.)

**Keywords:** sodium, end-stage kidney disease, peritoneal dialysis

## Abstract

Sodium overload is common in end-stage kidney disease (ESKD) and is associated with increased cardiovascular mortality that is traditionally considered a result of extracellular volume expansion. Recently, sodium storage was detected by Na23 magnetic resonance imaging in the interstitial tissue of the skin and other tissues. This amount of sodium is osmotically active, regulated by immune cells and the lymphatic system, escapes renal control, and, more importantly, is associated with salt-sensitive hypertension. In chronic kidney disease, the interstitial sodium storage increases as the glomerular filtration rate declines and is related to cardiovascular damage, regardless of the fluid overload. This sodium accumulation in the interstitial tissues becomes more significant in ESKD, especially in older and African American patients. The possible negative effects of interstitial sodium are still under study, though a higher sodium intake might induce abnormal structural and functional changes in the peritoneal wall. Interestingly, sodium stored in the interstial tissue is not unmodifiable, since it is removable by dialysis. Nevertheless, the sodium removal by peritoneal dialysis (PD) remains challenging, and new PD solutions are desirable. In this narrative review, we carried out an update on the pathophysiological mechanisms of volume-independent sodium toxicity and possible future strategies to improve sodium removal by PD.

## 1. Introduction

The kidneys play a pivotal role in the body’s sodium balance [1]; indeed, kidney failure is a well-known condition coupled with a remarkable sodium overload, which by inducing expansion of the extracellular volume (ECV), causes blood pressure (BP) elevation, left ventricular hypertrophy, and heart failure [2,3]. More interestingly, recent studies have highlighted that sodium accumulates in the interstitial tissue of the skin and in other tissue, leading to salt-sensitive hypertension regardless of the ECV expansion [4]. Hence, restoring the sodium body balance in end-stage kidney disease (ESKD) patients represents a cornerstone of dialytic treatment, but remains challenging [5].

This narrative review provides an update on the novel pathophysiological model of sodium toxicity in patients with chronic renal failure, and, in particular, in ESKD patients and looks at possible future strategies to improve sodium removal by PD.

## 2. Extrarenal Mechanisms for Sodium Body Balance and Hypertension

The classical two-compartment model for the body’s sodium balance assumes that the sodium intake is the primary determinant of ECV. According to this traditional model, in healthy humans in a steady state, the sodium intake is closely correlated to urinary sodium excretion. An abrupt increase in sodium intake, by enhancing plasma osmolarity, induces thirst and the secretion of ADH with a subsequent increase in intake and renal reabsorption of water, which in turn results in ECV expansion and body weight increase. After three to four days, a new steady state is achieved by increased sodium renal excretion, re-establishing the volume status. In this picture, sodium excretion is finely regulated by neuro-hormonal mechanisms (the renin–angiotensin–aldosterone system and sympathetic nervous system) activated by volume and pressure sensors (Figure 1a) [1]. This model of the body’s sodium balance has recently been challenged. Heer et al. showed that a high sodium intake was not associated with an increase in total body water or body weight but caused a relative fluid shift from the interstitial into the intravascular space [6]. Furthermore, a space-simulation study reported that in healthy individuals with a stable intake of nutrients, the sodium excretion changed periodically and, more important, independently of the sodium intake [7]. These findings have generated a novel hypothesis as to the internal sodium balance. According to the three-compartment model, an amount of sodium stored in the interstitial space is not osmotically inert, since by increasing local tonicity, it activates monocyte–phagocyte system (MPS) cells. Indeed, MPS cells sense enhanced tonicity in the interstitium by a tonicity enhancer binding protein (ton-EBP). Activation of the ton-EBP stimulates the production and release of interleukins (e.g., monocyte chemoattractant protein-1, MCP-1, IL-6) and the vascular endothelial growth factor-C (VEGF-C) which, causing local inflammation and vascular proliferation, induces sodium clearance by the lymphatic system (Figure 1b) [8]. Several experimental studies have demonstrated that salt-sensitive hypertension may be induced by blocking the Ton-EBP/VEGF-C pathway at various levels: Ton-EBP transcription, MPS cell recruitment, VEGF receptors, and lymph vessels [9,10,11]. Hence, interstial tissue could act as a sodium buffer on the part of the MPS cells and VEGF-C-dependent lympho-angiogenesis, so that a sodium overload may be the result of saturation (failure) of sodium interstitial storage. 

Furthermore, several experimental studies have also reported an abnormal activation of the adaptive immune system (Th1 and Th17 cells) as a result of higher sodium exposure, thus contributing to the development of hypertension. This suggests a more complex link between salt-induced inflammation and hypertension [12]. According to a recent working hypothesis, a higher salt intake stimulates the dendritic cells (DC) to produce reactive oxygen species via NADPH oxidase, which promotes the lipid peroxidation of fatty acids to isolevuglandins. These lipid oxidation derivates to lysine on proteins, which are in turn denatured, leading to the production of neoantigenic peptides, isolevuglandin adducts. These neoantigens activate the DCs to release proinflammatory cytokines, such as IL-6 and IL-1β. Furthermore, the DCs activated by isolevuglandin adducts stimulate the T cells to release IL-17, TNF-α, and IFN-γ, leading to hypertension and hypertension-damaged organ tissue (kidney, heart, vascular system) [13]. The gut microbiome has recently been added to this already complex scenario; thus, excess dietary salt might alter the gut microbiome, reducing *Lactobacillus* spp. and causing the activation of Th17 cells which migrate to the kidney and throughout the cardiovascular system, eventually leading to salt retention and dysfunction [14,15].

## 3. Volume-Independent Sodium Toxicity in End-Stage Kidney Disease

An optimal balance between intake and removal of sodium and water is currently considered as a major determinant of dialysis adequacy, even more than small solute clearances (urea in primis) [16]. Fluid overload is detected in over 50% of uremic patients treated by chronic dialysis [17,18] and contributes to their peculiar enhanced risk of cardiovascular morbidity and mortality [19,20]. However, the deleterious effects of sodium on the cardiovascular system in ESKD might also be independent of the ECV expansion. Indeed, the recent availability of Na^23^-MRI has enabled us to detect interstitial sodium storage in the skin and muscles and reveals new scenarios in the pathogenesis of sodium toxicity in ESKD. More specifically, clinical studies using Na^23^-MRI have shown that sodium stored in the interstitial skin tissue correlates with BP levels [21] and progressively increases as the GFR declines [22,23]. Interestingly, a cross-sectional study in a cohort of 99 CKD patients reported that skin sodium detected by Na^23^-MRI correlated with a greater left ventricular mass irrespective of the fluid overload and ambulatory BP [22]. This interstitial sodium concentration is even higher in the skin and in the muscles of ESKD patients as compared to the controls and CKD patients [23,24,25]. Furthermore, older and African American patients on maintenance dialysis exhibit a greater sodium content than do younger and non-African American ESKD patients [23,24]. Interestingly, in the study by Sahinoz et al., higher plasma inflammatory markers (IL-6 and C-reactive protein) correlated with an increased muscle and skin sodium content [24]. This correlation heralds an interesting link between interstitial sodium and chronic inflammation, as already reported in experimental studies [26,27,28]. Thus, in nephrectomized rats, exposure to two weeks of higher salt intake increased the levels of IL-6, CMP-1, and TNF-alfa in the dialysate. Notably, it was also associated with an increased expression of peritoneal Ton-EBP, which is the protein acting as a link between sodium and inflammation in interstitial tissue. Finally, they reported that increasing the sodium intake induced an increase in peritoneal permeability (D/P creatinine increase). Each of these observed abnormalities reverted to normal after the sodium intake was reduced [26]. One notes that the results of this study [26] were consistent with findings from an early experiment on Winster rats showing that exposure to a higher sodium intake was associated with the thickening of the peritoneal membrane and peritoneal fibrosis as well as causing an epithelial-to-mesenchymal transition [27]. These findings are also supported by a further experimental study, which showed that a higher salt intake was associated with macrophage infiltration, detectable not only in the heart and para-aortic tissues but also in the peritoneal wall [28]. 

### Sodium Removal by Peritoneal Dialysis

Recent insights into the negative effects of sodium stored in the skin interstitium have renewed interest in the dialytic strategies to remove sodium by dialysis [16,24,29,30,31]. Sodium stored in the interstitial tissue is by no means unmodifiable, given that a significant reduction in interstitial sodium has been documented after a hemodialysis session [29], while a higher ultrafiltration proves to be associated with lower skin sodium in patients on PD [24]. Furthermore, in hemodialysis patients, the use of low-sodium dialysate is associated with a lower sodium concentration in the skin and muscles [30]. Finally, in predialysis CKD patients, kidney transplantation caused tissue sodium accumulation to drop to the level of healthy controls [31].

Sodium is currently removed by convection in PD, because sodium removal by diffusion is generally very slight. Thus, sodium removal must be coupled with water removal by a convective mechanism (drag solvent) in order to become significant. A quantitative analysis in vivo, using two liters of glucose solution at different concentrations for 360 min dwells, estimated a sodium removal of −1.8 ± 26 mmol for 1.36%, 36.0 ± 21.0 mmol for 2.27%, and 70.5 ± 31.5 mmol for 3.86%, respectively [32]. Furthermore, the entity of the ultrafiltration (and sodium removal) depends on peritoneal permeability being significantly reduced in fast transporters [32]. Importantly, the UF does not correspond to sodium removal in Automated Peritoneal Dialysis, due to the greater removal of free water during short dwells, so that icodextrin needs to be used in the long dwells in order to remove the sodium overload, mostly in anuric patients and/or those with an unbalanced sodium intake [5]. Icodextrin is a polydispersed glucose polymer (mean molecular weight: 16,800 Da) that does not diffuse across the peritoneal membrane (reflection coefficient approaching 1.0 for high molecular weight fractions) unlike glucose solutions. Icodextrin increases the oncotic pressure within the peritoneal cavity, maintaining a colloid osmotic gradient across the peritoneal membrane, irrespective of the osmolarity (282 mOsm). During a dwell with icodextrin, a real convective sodium clearance occurs via the small pores and can be quantified at about 1 g of sodium chloride (15 mmol) for every 100 mL of ultrafiltration obtained. It is evident that sodium sieving cannot occur because aquaporins are not involved and, consequently, there is no free water transport [16]. 

## 4. Alternative Strategies to Remove Sodium by Peritoneal Dialysis

The sodium concentration in standard PD solutions (132/134 mmol/L) is slightly different from blood concentration, thus lowering sodium in the dialysate has been proposed as a potential strategy to increase the sodium gradient between blood and dialysate and thus improve sodium removal; however, no benefit associated with the use of low-sodium PD solutions has yet been reported. The first studies using ultralow sodium (approximately 100 mmol/L) solutions dated from the beginning of the 1980s; these studies showed that ultralow sodium PD solutions were able to improve sodium removal up to three times more than conventional fluid but were associated with UF failure (due to a reduction of solution osmolality) and a higher risk of hyponatremia [33,34,35]. Furthermore, in a small, nonrandomized study, Davies et al. compared the effects on sodium removal and UF found with using uncompensated ultralow sodium (102 mmol/L) vs. compensated low sodium solutions (115 mmol/L); the authors reported that improvements in BP, thirst, and fluid status were only achieved when increasing the diffusive component of sodium removal whilst maintaining ultrafiltration by glucose-compensated solutions [36]. More recently, Rutkowski et al., using a PD solution with a reduced sodium content of 125 mmol/L and unchanged levels of glucose, reported a decrease in BP levels, with less risk of UF loss and hyponatremia; however, the trial failed the primary endpoint (dialysis adequacy) vs. the control (conventional PD solutions) [37]. A post hoc analysis concluded that a greater effect on BP levels was evident in patients with a residual eGFR <6 mL/min/1.73 m^2^ [38]. One trial recently reported that a glucose-compensated, low-Na PD solution (112 mmol/L Na and 2% glucose), as compared to a standard-Na solution (133 mmol/L Na and 1.5% glucose), was not associated with an improvement of BP in 123 hypertensive patients on PD [39]. All studies evaluating low-sodium PD solutions thus suggest that a mild but persistent (in all PD dwells) reduction in the sodium content of PD bags might help improve BP control with no need to increase the glucose concentration to maintain ultrafiltration [40]. Accordingly, we recently used an uncompensated glucose 130 mmol/L sodium PD solution to treat two hypertensive PD patients with no sign of volume overload. After six months of a 130 sodium–glucose solution in all dwells, we registered a significant reduction in daytime and night-time BP, with a dependent restoration of the circadian BP rhythm and no difference in body weight, ultrafiltration, or residual kidney function [41].

An alternative strategy to enhance sodium removal might be a mixed solution of crystalloid and colloid PD. In a small study using a bimodal solution containing glucose (2.6%) and icodextrin (6.8%) and a low-sodium solution (121 mmol/L) in automated PD, Freida et al. showed a significant rise in sodium and fluid removal (up to 2.5-fold) as compared to icodextrin alone (7.5%) [42]; however, this strategy has not been replicated in larger studies.

Finally, promising results may come from the new PD solutions containing L-carnitine and xylitol as osmotic agents [43,44,45,46,47]. Indeed, the use of more biocompatible crystalloid agents [43,44,45,46] might enable compensated low-sodium PD solutions to be formulated. The initial clinical experience using L-carnitine and xylitol solutions in ten CAPD patients over four consecutive weeks proved safe and well tolerated [47]; however, larger trials designed ad hoc are needed to verify the efficacy and safety of these new strategies in ESKD patients treated by PD.

## 5. Conclusions

Sodium overload in ESKD patients is associated with an increased CV risk. Sodium toxicity is due not only to the expansion of the ECV but also to inflammatory mechanisms that are independent of volume, being primarily mediated by the immune cells. Sodium storage increases in the interstitial tissues as the GFR declines and is removable by dialysis in ESKD patients. Reducing the sodium overload is therefore mandatory when treating hypertensive PD patients, regardless of the fluid status. Thus, alternative PD solutions are desirable that are geared to improving the internal sodium balance of these patients

## Figures and Tables

**Figure 1 ijms-22-12804-f001:**
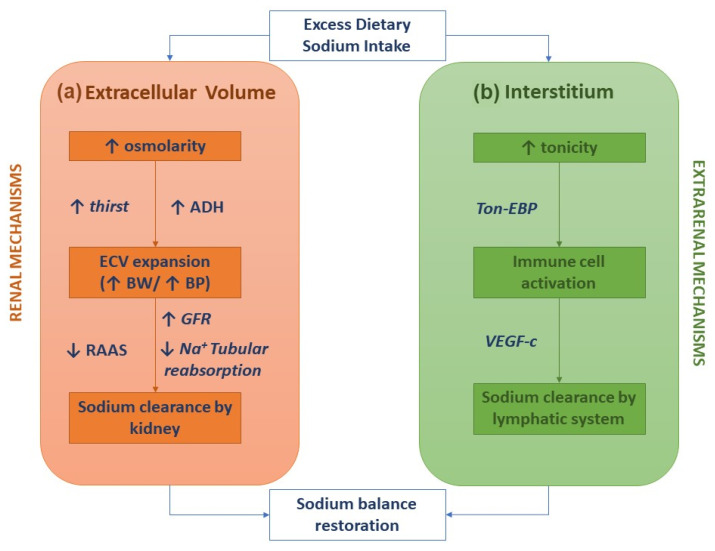
Sodium homeostasis according to the traditional kidney-regulated model (**a**) and the novel immune-regulated model (**b**).

## Data Availability

Not applicable.

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
