# Peer review of "Volume-Independent Sodium Toxicity in Peritoneal Dialysis: New Insights from Bench to Bed"

_ijms, 2021, doi:10.3390/ijms222312804_

Round 1

Reviewer 1 Report

The paper by Borrelli et al., is a comprehensive and properly focused mini-review that addresses the key aspects of sodium overload toxicity. The manuscript properly summarize the mechanism of volume-independent sodium toxicity highlighting the potential role of peritoneal dialysis as strategy to remove sodium.  Thus, the topic is of a significant importance, the review is well-written, up-to-date and worthy of publication.
Only minor concerns need to be addressed by the authors:

  • The sentence from line 135 to 138 should be better described. Some reader could not know the difference between Automated Peritoneal Dialysis and Continuous Ambulatory Peritoneal Dialysis (even if not mentioned here) or could not know what is icodextrin and why it is used.

Author Response

We are very grateful to Reviewer 1 for these positive comments. 

As suggested by Reviewer 1, we have clarified the issue on APD and icodextrin use.

Please find the new version of the manuscript with the revision of the English language.

Reviewer 2 Report

The work concerns a very significant problem of sodium toxicity in peritoneal dialysis patients. Sodium consumption in industrialized countries remains high, but a healthy person has mechanisms to balance sodium and chloride in average consumption. Renal failure leads to ineffective sodium excretion and, consequently, the development of hypertension, heart failure and cardiovascular death. Also among patients on peritoneal dialysis, the main cause of death is cardiovascular disease, which can be caused by, inter alia, poorly controlled arterial hypertension. One of the reasons for this may be a lack of sufficient sodium removal, especially with excessive sodium consumption. Hence the interest in this problem. The article presents in an accessible way the current views on the pathomechanisms of sodium retention and the possibility of its removal by peritoneal dialysis.

Author Response

We are very grateful to Reviewer 2 for the positive comments.

As suggested by the reviewer, the manuscript has been reviewed by an English native.

This manuscript is a resubmission of an earlier submission. The following is a list of the peer review reports and author responses from that submission.

Round 1

Reviewer 1 Report

Overall an interesting review highlightling the complexity of treating patients with ESRD. However the link between PD, immunefunction and different solutes in driving the gradient is needed. My suggestion is that authors review their aim and scope of manuscript. What is the focus?

  1. The models of sodium clearance in ESRD
  2. the immuneregulation in hypertension and its role in ESRD
  3. New ways of utilising the peritoneum for dialysis

If all 3 then the transition between each needs to made clear.

Further:

be consistent between kidney and renal

Introduction and abstract especially need to be reviewed for language.

It is Ångström (angstrom) not Armstrong for poresize.

Reference is needed for statement  "that continued use of icodextrin does not cause adverse effects" (line 194-5).

Regarding peritonitis as a factor for spontaneously changing permeability (line 222) then authors should be careful with phrasing given both the underlying cause for peritonitis and the immune effects driving the changed permeability.

Reviewer 2 Report

The authors provide a narrative piece regarding sodium and water balance in patients under peritoneal dialysis. Overall, the structure of this narrative review appears rather unorganized, without a clear manuscript flow demonstrated. Several concerns are outlined below.

  1. Much of the review content is mostly known from the existing literature (ex. sections “functional anatomy of peritoneum”, “sodium removal by peritoneal dialysis”). Therefore, these sections may be deemed unnecessary and be removed.
  2. The connection between sodium/water balance in peritoneal dialysis patients and extrarenal sodium/water reservoir is unclear, and the latter section simply jumps out from nowhere. It is also likely that the “extrarenal sodium/water reservoir” section can be removed, since this essentially plays no role in the content of this narrative review.
  3. Last but not least, sodium toxicity in patients with end-stage renal disease and those under peritoneal dialysis and relevant content summary has already been summarized in recent reviews (ex. J Nephrol 2020;33:59-68, Kidney Res Clin Pract 2021;40:135-42). There seems to be no need to repeatedly addressing this issue in this manuscript. From this perspective, it is felt that most of the content in this narrative piece is either unnecessary or available elsewhere.